# Combination of Ad-SGE-REIC and bevacizumab modulates glioma progression by suppressing tumor invasion and angiogenesis

**Yasuhiko Hattori**[1], **Kazuhiko Kurozumi**[2]*, **Yoshihiro Otani**[1], **Atsuhito Uneda**[1], **Nobushige Tsuboi**[1], **Keigo Makino**[1], **Shuichiro Hirano**[1], **Kentaro Fujii**[1], **Yusuke Tomita**[1], **Tetsuo Oka**[1], **Yuji Matsumoto**[1], **Yosuke Shimazu**[1], **Hiroyuki Michiue**[3], **Hiromi Kumon**[4], **Isao Date**[1]

1 Department of Neurological Surgery, Okayama University Graduate School of Medicine, Dentistry and Pharmaceutical Sciences, Okayama, Japan, 2 Department of Neurosurgery, Hamamatsu University School of Medicine, Shizuoka, Japan, 3 Neutron Therapy Research Center, Okayama University, Okayama, Japan, 4 Innovation Center Okayama for Nanobio-targeted Therapy, Okayama University, Okayama, Japan

* kurozu20@hama-med.ac.jp

**Data Availability Statement:** **PA at Accept: Please follow up with authors to ensure data are publicly available in GEO database** All microarrya

## Abstract

Reduced expression in immortalized cells/Dickkopf-3 (REIC/Dkk-3) is a tumor suppressor and its overexpression has been shown to exert anti-tumor effects as a therapeutic target gene in many human cancers. Recently, we demonstrated the anti-glioma effects of an adenoviral vector carrying REIC/Dkk-3 with the super gene expression system (Ad-SGE-REIC). Anti-vascular endothelial growth factor treatments such as bevacizumab have demonstrated convincing therapeutic advantage in patients with glioblastoma. However, bevacizumab did not improve overall survival in patients with newly diagnosed glioblastoma. In this study, we examined the effects of Ad-SGE-REIC on glioma treated with bevacizumab. Ad-SGE-REIC treatment resulted in a significant reduction in the number of invasion cells treated with bevacizumab. Western blot analyses revealed the increased expression of several endoplasmic reticulum stress markers in cells treated with both bevacizumab and Ad-SGE-REIC, as well as decreased β-catenin protein levels. In malignant glioma mouse models, overall survival was extended in the combination therapy group. These results suggest that the combination therapy of Ad-SGE-REIC and bevacizumab exerts anti-glioma effects by suppressing the angiogenesis and invasion of tumors. Combined Ad-SGE-REIC and bevacizumab might be a promising strategy for the treatment of malignant glioma.

## Introduction

Glioma is the most common type of primary brain tumor, and malignant gliomas are aggressive intracranial neoplasms in humans [1, 2]. The key characteristics of malignant gliomas are proliferation, angiogenesis, and invasion. Effective chemotherapeutic or molecular-targeted agents for malignant glioma have yet to be developed. Owing to the high resistance of malignant glioma to current therapies, new therapeutic agents are needed immediately.

files are available from the Gene Expression Omnibus (accession number GSE205928).

**Funding:** This study was supported by Japan Society for the Promotion of Science(JSPS) KAKENHI Grant Numbers 26462182 and 17K10865 (K. Kurozumi).

**Competing interests:** H. Kumon is the Chief Scientific Officer of Momotaro-Gene Inc. He demonstrated the utility of the agent and also owns stocks in Momotaro-Gene Inc. The other authors have no other relevant affiliations or financial involvement with any organization or entity with a financial interest in or financial conflict with the subject matter or materials discussed in the manuscript apart from those disclosed.

Reduced expression in immortalized cells/Dickkopf-3 (REIC/Dkk-3) is a tumor suppressor gene, and expression of the *REIC/Dkk-3* gene was shown to be decreased in various human tumors, including glioma [3–6]. The adenovirus REIC vector with the super gene expression system (Ad-SGE-REIC) was developed to overexpress REIC/Dkk-3, and it demonstrated *in vitro* and *in vivo* anti-tumor effects on malignant glioma [7]. We also observed reduced expression of β-catenin [7], the key nuclear effector of the Wnt signaling pathway that regulates cell fate [8], in glioma cells infected with Ad-SGE-REIC. As a result, the Good Manufacturing Practice product of Ad-SGE-REIC was developed. After consultations with the Japanese Pharmaceuticals and Medical Devices Agency (PMDA), we submitted a clinical trial notification for a phase I/IIa study of Ad-SGE-REIC for the treatment of recurrent malignant brain tumors in 2019 [9].

In addition to the function of REIC/Dkk-3 as a tumor suppressor, previous studies showed that REIC/Dkk-3 may also exhibit angiogenic activity during embryogenesis and tumorigenesis [10]. One report revealed that Dkk-3 enhances the phosphorylation of Smad1/5/8 and recruits Smad4 to the VEGF gene promoter, suggesting that REIC/Dkk-3 activates the transcription of VEGF and induces angiogenesis [11]. Bevacizumab is a humanized monoclonal antibody that targets VEGF and was first approved in the United States for glioblastoma in 2009. In Japan, bevacizumab is approved for newly diagnosed and recurrent malignant glioma. While two phase III trials reported that bevacizumab did not improve the survival of patients with newly diagnosed glioblastoma [12, 13], improved progression-free survival and maintenance of baseline quality of life and performance status were observed in the bevacizumab group in the AVAGlio study [12]. Furthermore, recent studies suggested the additional efficacy of bevacizumab in conjunction with other therapies, including immunotherapies, by decreasing VEGF [14]. Thus, we hypothesized that bevacizumab may decrease REIC/Dkk-3-mediated VEGF upregulation and angiogenesis, thus increasing the efficacy of Ad-SGE-REIC.

In this study, we evaluated the combination treatment of Ad-SGE-REIC and bevacizumab on glioma. Bevacizumab suppressed tumor angiogenesis, and Ad-SGE-REIC reduced bevacizumab-induced glioma invasion. We also observed the decreased activation of the β-catenin signaling pathways in glioma cells treated with combined Ad-SGE-REIC and bevacizumab.

## Materials and methods

### Cell lines, drugs, and adenovirus vector

Human GBM cell lines U87ΔEGFR and U251MG were prepared and maintained as described previously [15]. These cell lines were provided by Dr. Balveen Kaur (The University of Texas Health Science Center at Houston, Houston, TX, USA). Human glioblastoma-derived cell lines MGG8 and MGG23 were provided by Dr. Hiroaki Wakimoto of Massachusetts General Hospital (Boston, MA, USA) and cultured as previously described [16, 17]. Human umbilical vein endothelial cells (HUVECs) were purchased from Lonza (Basel, Switzerland). Mycoplasma was negative in all cell lines. Bevacizumab was purchased from Genentech (San Francisco, CA, USA)/Roche (Basel, Switzerland)/Chugai Pharmaceutical Co. (Tokyo, Japan). Ad-SGE-REIC was provided as described previously [7, 18].

### Cytotoxicity assay

Cells were cultured in flat-bottomed six-well dishes at a concentration of $4.0 \times 10^5$ cells/well. The cells were treated with phosphate-buffered saline (PBS) (Control), Ad-SGE-REIC at an MOI of 30 (SGE), 1 mg/mL bevacizumab (Bev), or both Ad-SGE-REIC and bevacizumab (SGE + Bev). After 0, 24, 48, 72, and 96 h, cell viability was examined. The number of living

cells attached to the bottom of each culture well and floating in medium was determined by trypan blue staining and counted using a LUNA-II automated cell counter (Logos Biosystems, Anyang, South Korea).

## Sphere-forming assay

MGG8 and MGG23 cells were dissociated by trypsin and plated in Neurobasal medium (Invitrogen, Waltham, MA, USA) supplemented with 3 mmol/L l-Glutamine (Mediatech, Laval, QC, Canada), 1× B27 supplement without vitamin A (Life Technologies, Carlsbad, CA, USA), 0.5× N2 supplement (Life Technologies), 20 ng/mL recombinant human EGF (R&D Systems, Minneapolis, MN, USA), 20 ng/mL recombinant human FGF2 (Peprotech, East Windsor, NJ, USA), and 0.5× penicillin G/streptomycin sulfate/amphotericin B complex (Mediatech). A total of $2 \times 10^3$ cells/well were plated in 24-well ultra-low attachment plates (Corning, Lowell, MA, USA) [19]. The cells were infected with Ad-SGE-REIC at an MOI of 30 or treated with 1 mg/mL bevacizumab. The number of spheres that formed after 10 days was counted.

## Western blot analysis

Total cell protein was extracted from cells using ice-cold lysis buffer (20 mM Tris pH 7.5, 150 mM NaCl, 1 mM EDTA, 1 mM EGTA, 1.0% Triton X-100, 1 tablet/10 cc buffer of PhosSTOP [Roche Applied Science, Mannheim, Germany], and protease inhibitor cocktail) and protein concentration was quantified using the Bradford method [20]. Equal amounts of protein (20 μg) were separated by sodium dodecyl sulphate–polyacrylamide gel electrophoresis (SDS-PAGE) and then blotting was performed as described previously [13, 18]. The primary antibodies included mouse anti-human REIC/Dkk-3 (Momotaro-Gene Inc., Okayama, Japan), rabbit anti-pIRE1α (Novus Biologicals, Littleton, CO, USA), rabbit anti-human BiP, rabbit anti-human β-catenin, TBP (Cell Signaling Technology, Danvers, MA, USA), and mouse anti-human β-actin antibody (Sigma, St. Louis, MO, USA); all primary antibodies were diluted 1:2000 in Can Get Signal® (Toyobo Co., Ltd., Osaka, Japan). The secondary antibody horseradish peroxidase-conjugated anti-mouse or anti-rabbit IgG (Cell Signaling Technology) was diluted 1:5000 in 1% skim milk [7]. We quantified band densities using Image J (ver,1.53r).

## Tube formation assay

EGM™ -2 Endothelial Growth Medium-2 BulletKit™, HBSS, Trypsin/EDTA, and Trypsin neutralizing solution (Lonza) were used following the manufacturer's instructions. Matrigel® (Corning) was thawed overnight in the refrigerator at 4˚C; a 96-well culture plate and pipetting tips were pre-cooled with Matrigel in the refrigerator overnight and kept on ice during the coating process. Matrigel was added to 96-well culture plates (50 μL/well), and the plate was incubated at 37˚C for 30 min. HUVECs were plated in the plates ($2.0 \times 10^4$) with conditioned medium obtained by cultivating U251MG cells in a cytotoxicity assay. After 24 h, the cells were observed under a microscope (BZ-8000; Keyence, Osaka, Japan). The total tube length in each well was calculated under high magnification (×20).

## Transwell migration assay

*In vitro* migration assays were performed using a 24-well plate and ThinCert (8 μm pore, 24-well format, Greiner Bio-one; Kremsmunster, Austria) following the manufacturer's instructions. We prepared U87ΔEGFR cells as described previously [11], and cells were treated with PBS, bevacizumab, and/or Ad-SGE-REIC as in the cytotoxicity assay. After the treatments, the conditioned medium (CM) and cells of each treatment group were collected. The

upper chamber was filled with $2.0 \times 10^5$ U87ΔEGFR cells in 200 μL CM; the lower chamber was filled with DMEM with 10% fetal bovine serum (FBS) as a chemoattractant with or without 1 mg/mL bevacizumab. After a 24 h incubation, non-invading cells were scraped from the top compartment. The insert filters were stained with 5% Giemsa solution, and the number of invading cells was counted on the lower surface of the filter [14, 21].

### Ethics and animal use statement

This study was conducted in strict accordance with the recommendations in the Guide for the Care and Use of Laboratory Animals in Japan. All procedures and animal protocols were in accordance with the guidelines and approved by the Committee on the Ethics of Animal Experimentation at Okayama University (Permit No. OKU-2016554).

### Brain xenografts

Athymic mice (BALB/c-nu/nu) were obtained from CLEA Japan, Inc. (Tokyo, Japan). They were housed in groups of up to five mice in single cages at 23°C under a 12–12 light/dark cycle with lights on at 8:00 AM. They accessed to food and water ad libitum. We prepared $2.0 \times 10^5$ of U87ΔEGFR cells as described previously [22]. Mice were anesthetized with an intraperitoneal (ip) injection of ketamine (2.7–3.0 mg/100 g) and xylazine (0.36–4.0 mg/100 g). Cells (2 μL) were injected into the right frontal lobe of mice (3 mm lateral and 1 mm anterior to the bregma at a depth of 3 mm), as described previously. Thirty-six mice were randomly divided into the control group (PBS), bevacizumab group, SGE group (Ad-SGE-REIC) and SGE+bevacizumab group, each of which contained 9 mice. PBS or bevacizumab (10 mg/kg) was intraperitoneally administered two times per week, starting on day 5 after tumor cell implantation. At 7 days after tumor inoculation, all mice bearing brain tumors were reanesthetized and stereotactically injected with Ad-SGE-REIC or PBS at the tumor inoculation site using the same coordinates. We assessed the survival time of the U87ΔEGFR mouse glioma model using Kaplan–Meier survival analysis.

Thirty-two mice were randomly divided into the same four groups, each of which contained 8 mice. Eighteen days after tumor implantation, following four administrations of PBS or bevacizumab and stereotactically injected with Ad-SGE-REIC or PBS, mice were anesthetized with an ip injection of ketamine and xylazine and euthanized by exsanguination. The brains were obtained for histologic examination (n = 5 each group) or Microarray analysis (n = 3 each group).

In all animal experiments, body weight, intake of food and water, and neurological findings were monitored for all mice every day throughout the whole experimental period. If they had hemiplegia, convulsions, ataxia, or body weight loss $\geq$ 20% from baseline body weight during the observation time, the mice were immediately euthanized the same as above.

### Immunohistochemistry

Immunohistochemistry was performed using the avidin-biotin-peroxidase complex method (Ultrasensitive; MaiXin, Fuzhou, China). CD31 mouse monoclonal antibody (1:300 dilution; Abcam, Inc., Cambridge, UK), anti-human leukocyte antigen mAb (1:100 dilution; Abcam Inc.), and mouse immunoglobulin (a negative control) were used as previously described [16]. Hematoxylin was used for counterstaining.

### Microarray analysis

Orthotopic U87ΔEGFR xenograft mouse models treated with the control, Ad-SGE-REIC, or combination of bevacizumab and Ad-SGE-REIC were euthanized 18 days after tumor

implantation (n = 3/treatment). Approximately 40 mg of brain tumor samples were excised cleanly from each mouse, and RNA was extracted using TRIzol (Life Technologies) and an RNeasy Mini Kit (Qiagen, Venlo, Netherlands). Samples were analyzed using the SurePrint G3 Mouse GE Microarray 8x60K Ver2.0 (Agilent, Tokyo, Japan). The microarray analyses were conducted by DNA Chip Research Inc. (Tokyo, Japan). Amplification and biotin labeling of fragmented cDNA was performed using the NuGEN Ovation Pico WTA System V2 (NuGEN Technologies Inc., San Carlos, CA, USA). Labeled probes were hybridized to the Agilent Sure-Print G3 Mouse Gene Expression 8x60K and scanned. Expression data were extracted from image files produced on an Agilent microarray scanner. The scanned images were analyzed with Agilent Feature Extraction 12.1.1.1. Genechip analysis was performed using GeneSpring GX 14.9.1 software (Agilent). The median shift normalization to the 75th percentile and baseline transformation using the median of all samples was applied [23].

A gene was defined as upregulated when the Ad-SGE-REIC monotherapy/control or combination therapy/Ad-SGE-REIC monotherapy average intensity ratio was >4.0 and downregulated when the Ad-SGE-REIC monotherapy/control or combination therapy/Ad-SGE-REIC monotherapy ratio was <0.25. We performed pathway analysis on upregulated and downregulated genes using Microarray Data Analysis ToolVer3.2 (Filgen, Inc.). The data were extracted using the following criteria: Z score > 0 and $p$ value < 0.05 [22, 24]. All data are deposited in the Gene Expression Omnibus (accession number GSE205928).

### Statistical analysis

The Student's $t$-test, Mann–Whitney U test, and analysis of variance (ANOVA) were used to test for statistical significance. Data are presented as the mean ± standard deviation and standard error. $p < 0.05$ was considered to indicate statistical significance. All statistical analyses were performed using SPSS statistical software, version 20 (SPSS, Inc., Chicago, IL, USA).

## Results

### Cytotoxic effect of combination therapy with bevacizumab and Ad-SGE-REIC

To investigate the effect of Ad-SGE-REIC and bevacizumab in glioma cells, established glioma cells (U87ΔEGFR and U251MG) and glioma stem cells (MGG8 and MGG23 cells) were incubated with bevacizumab and Ad-SGE-REIC, alone or in combination. Consistent with our previous data [7], treatment of U87ΔEGFR and U251MG cells with Ad-SGE-REIC decreased cell viability compared with the control in a time-dependent manner (Fig 1A and 1B), with a significant decrease in the viability of cells treated with Ad-SGE-REIC compared with the control after 72 h (U87ΔEGFR, $p < 0.05$; U251MG, $p < 0.005$). After treatment with Ad-SGE-REIC, U87ΔEGFR cells aggregated and floated, whereas MGG8 and MGG23 cells were dissociated and remained adherent to the dishes (Fig 1C and 1D). There was a significant decrease in the viability of MGG8 and MGG23 cells treated with Ad-SGE-REIC compared with PBS at 10 days (MGG23, $p < 0.05$; MGG8, $p < 0.005$) (Fig 1E and 1F). Bevacizumab had no cytotoxic effect against glioma cells and did not impact the cytotoxicity of Ad-SGE-REIC against glioma cells.

### Anti-angiogenic effect of bevacizumab in the combination treatment

Next, to investigate the anti-angiogenic effect of bevacizumab, tube formation assays were performed using HUVECs. The CM of malignant glioma cells infected by Ad-SGE-REIC was centrifuged and filtrated to eliminate virus and cell debris. When HUVECs were cultured in the presence of VEGF, efficient tube formation was observed (Fig 2A and 2B). Treatment with

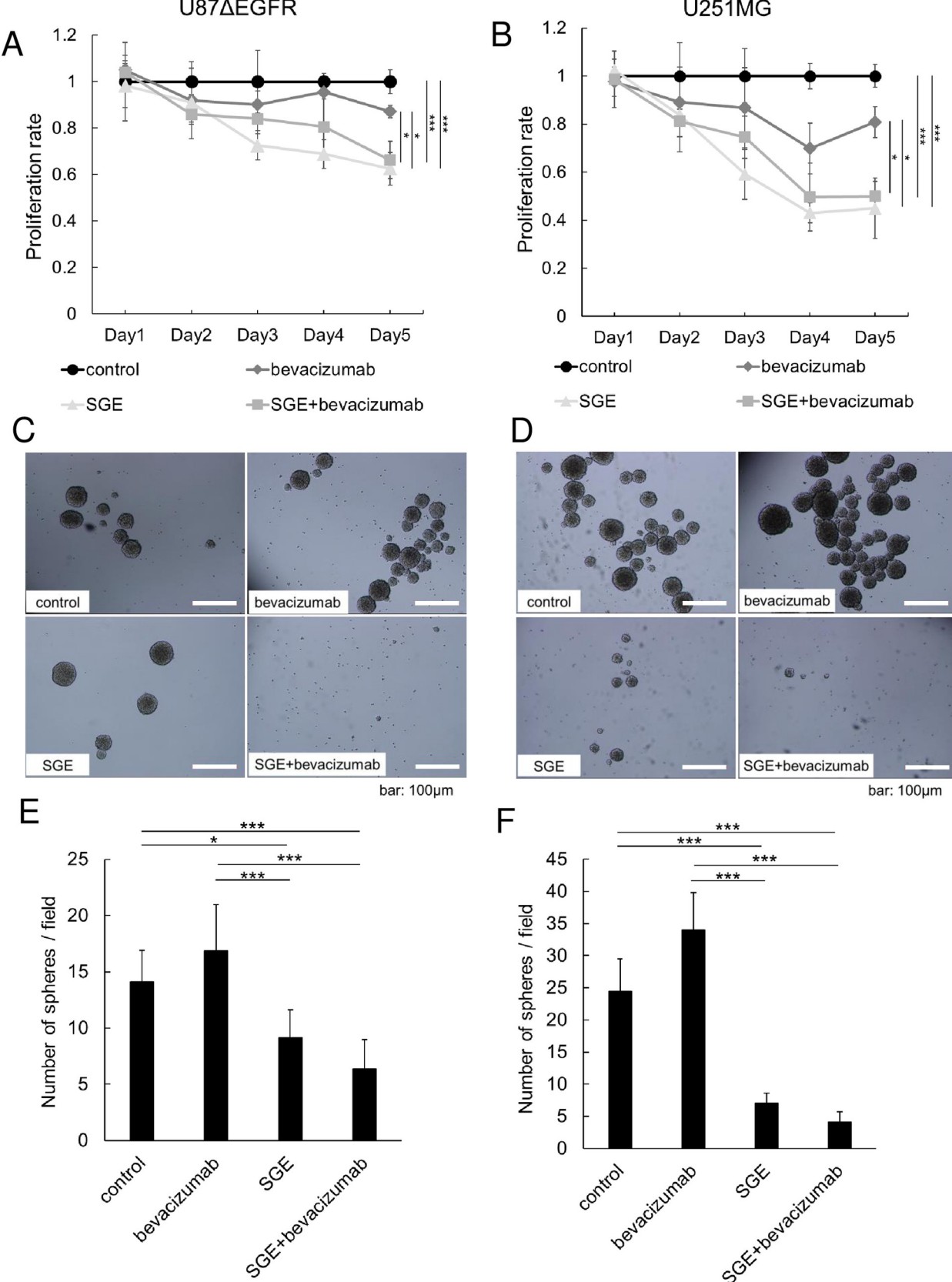

bar: 100μm

bar: 100μm

**Fig 1. Cytotoxic effect of Ad-SGE-REIC, bevacizumab, and the combination on glioma cell lines.** U87ΔEGFR (A) and U251MG (B) glioma cells were treated with saline as a control or bevacizumab (1 mg/ml) and infected with saline or Ad-SGE-REIC (SGE; MOI of 30). Cell viability was examined every 24 h after infection. The number of living cells attached to the bottom of each culture well and floating in the medium was determined by trypan blue staining. Data shown are the proportion of viable cells relative to control cells. Values are the mean ± standard deviation (SD) from five independent experiments. Statistical significance was calculated by one-way analysis of variance (ANOVA) with Tukey's *post hoc* test. Representative images from the sphere formation assay (C, D). The number of spheres formed from $10^4$ cells was counted. Data are expressed as the mean ± SD from three independent experiments(C, E: MGG8 cells, D, F: MGG23 cells). $^*p < 0.05$, $^{**}p < 0.01$, and $^{***}p < 0.005$ compared with the indicated groups.

bevacizumab efficiently inhibited tube formation compared with the control. In contrast, Ad-SGE-REIC CM increased the tube length in HUVECs. Notably, co-treatment of bevacizumab with Ad-SGE-REIC CM significantly decreased tube length compared with HUVECs treated with Ad-SGE-REIC CM alone.

## Ad-SGE-REIC-infected glioma cells showed inhibited migration activity *in vitro*

We next examined the effects of bevacizumab and Ad-SGE-REIC on glioma cell migration. Glioma cells treated with bevacizumab and/or the CM of Ad-SGE-REIC were seeded into upper chambers of Transwell chambers, and the cells that migrated through the membrane were counted 24 h later (Fig 2C). As we previously reported, bevacizumab induced glioma cell migration [21, 22, 25]. Ad-SGE-REIC significantly reduced the number of migrated cells of U87ΔEGFR compared with the control (Fig 2D). Furthermore, the number of migrated cells treated with bevacizumab was reduced by co-treatment with Ad-SGE-REIC.

## Endoplasmic reticulum (ER) stress and β-catenin degradation by combination of Ad-SGE-REIC and bevacizumab in glioma cells

We previously demonstrated the increased expressions of ER stress markers in U87Δ EGFR glioma cells treated with Ad-SGE-REIC. Thus, we next evaluated how combining bevacizumab affects ER stress. Western blot analysis indicated the increased expressions of ER stress markers, Bip and phosphorylated IRE1α in U87ΔEGFR and U251MG cells treated with the combination treatment compared with cells treated with either Ad-SGE-REIC or bevacizumab cells. (Fig 3A).

The Wnt signaling pathway regulates cell migration through the inhibition of the proteasome-dependent proteolysis of β-catenin [8]. Therefore, we evaluated the impact of the Ad-SGE-REIC and bevacizumab combination treatment on β-catenin expression in glioma cells (Fig 3B). The results showed that β-catenin protein levels were more potently reduced by the combination treatment compared with levels in cells treated with Ad-SGE-REIC or bevacizumab as well as controls in U87ΔEGFR and U251MG cells.

## Anti-tumor effect of the combination therapy with bevacizumab and Ad-SGE-REIC in xenograft mice

We evaluated the potential antitumor effect of the combination therapy of bevacizumab and Ad-SGE-REIC on mice harboring intracranial U87ΔEGFR glioma cells (Fig 4A). We compared mice bearing U87ΔEGFR glioma cells treated with saline, bevacizumab at 10 mg/kg, Ad-SGE-REIC at $3.6 \times 10^7$ pfu, and Ad-SGE-REIC at $3.6 \times 10^7$ pfu and bevacizumab at 10 mg/kg (Fig 4B). Control mice treated with PBS had a median survival of 14 days after tumor cell implantation. Mice treated with Ad-SGE-REIC had a median survival of 19 days after tumor cell implantation, and mice treated with bevacizumab had a median survival of 19 days after tumor cell implantation. Notably, mice treated with the bevacizumab and Ad-SGE-REIC

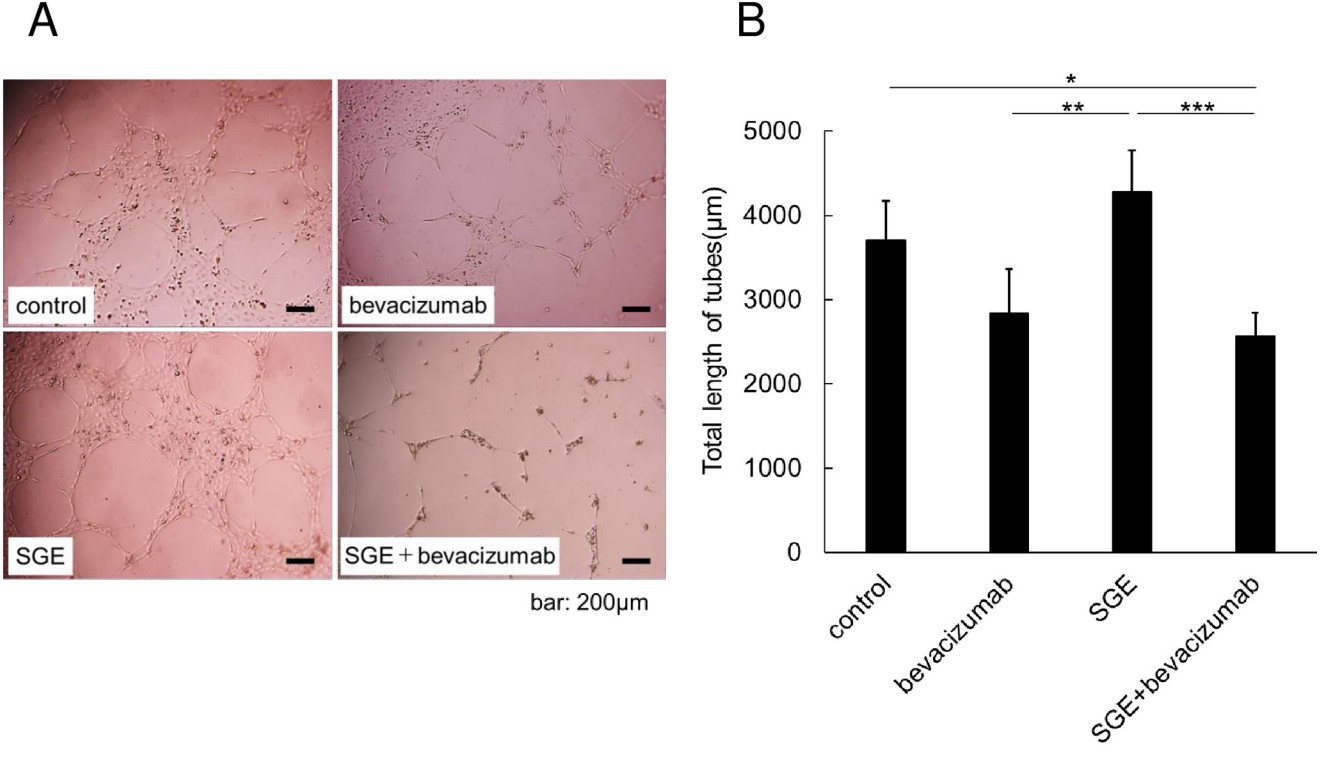

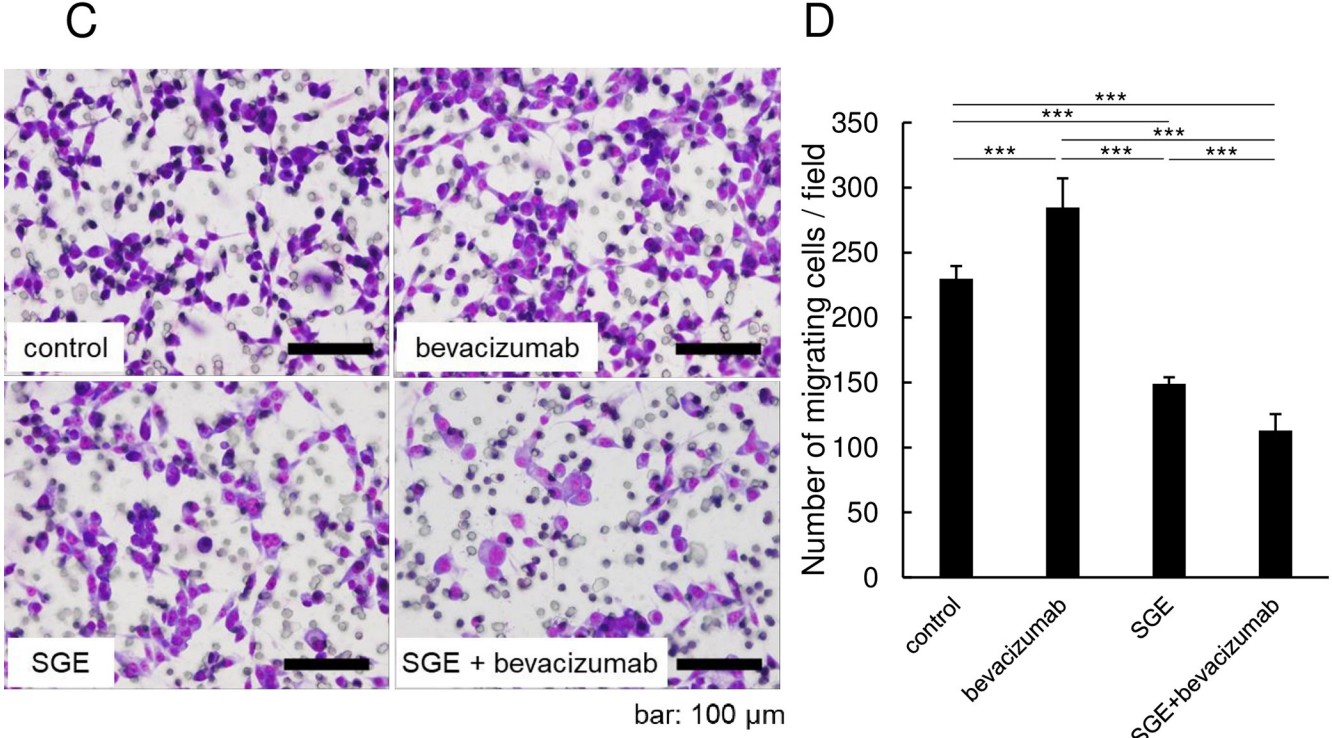

**Fig 2. Inhibition of endothelial cell tube formation.** Human umbilical vein endothelial cells (HUVECs) were treated with medium containing vascular endothelial growth factor (VEGF)-A along with Ad-SGE-REIC and/or bevacizumab and then assayed by a KURABO Angiogenesis Kit. (A) Representative

images of tube formation of HUVECs. Control: VEGF-A (10 ng/ml); SGE: VEGF-A/Ad-SGE-REIC ($5 \times 10^2$pfu); bevacizumab: VEGF-bevacizumab (0.1 mM); and SGE+bevacizumab: VEGF-A/ Ad-SGE-REIC/bevacizumab. (B) Data shown are the total length of endothelial cell tube formation. Significant reduction in the tube formation of HUVECs treated with combined Ad-SGE-REIC and bevacizumab compared with Ad-SGE-REIC or bevacizumab monotherapy. (C) Representative images from the two-chamber invasion assay. Glioma cell lines were incubated with conditioned medium (CM) derived from glioma cells treated with Ad-SGE-REIC. Cells were treated with bevacizumab. (D) Migrated cells were counted 24 h after treatment. Data shown are the number of invading cells relative to those treated with saline as a control. Values are the mean ± standard deviation (SD) from five independent experiments. Statistical significance was calculated by one-way analysis of variance (ANOVA) with Tukey's *post hoc* test. $^*p < 0.05$, $^{**}p < 0.01$, and $^{***}p < 0.005$ compared with the indicated groups.

combination had a median survival of 29 days, which was significantly longer than mice treated with PBS, Ad-SGE-REIC alone, or bevacizumab alone ($p < 0.001$, $p = 0.011$, and $p = 0.027$, respectively).

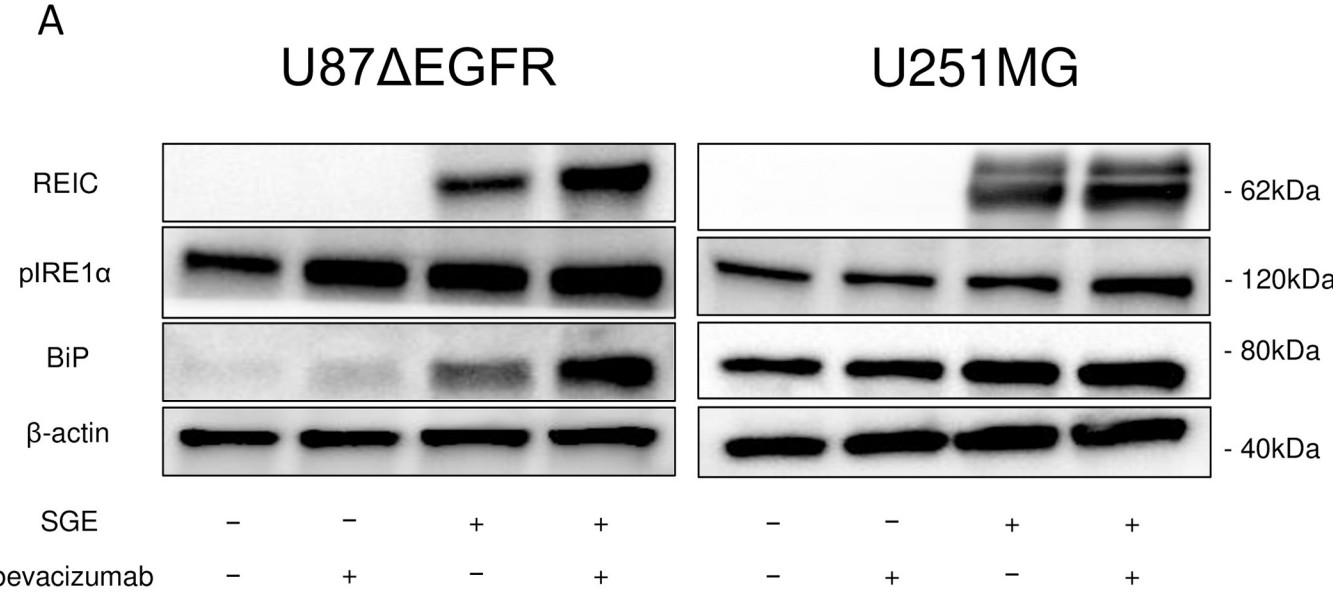

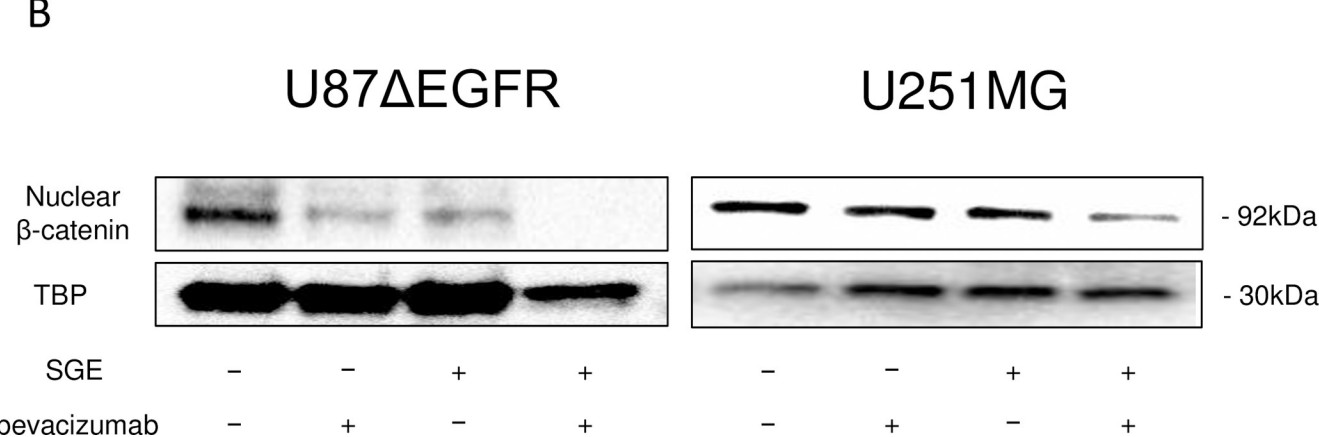

**Fig 3. ER stress in U87ΔEGFR glioma cells after treatment with Ad-SGE-REIC.** (A) U87Δ EGFR or U251MG cells were infected with Ad-SGE-REIC at a MOI of 10. Immunoblot analysis showed that levels of REIC, BiP, and phosphorylated IRE1$\alpha$ were increased following treatment with Ad-SGE-REIC. Moreover, protein levels increased in the combination treatment group. (B) A reduction in β-catenin expression occurred in parallel with increased expression of REIC/Dkk-3 (n = 4). Moreover, protein levels decreased in the combination treatment group.

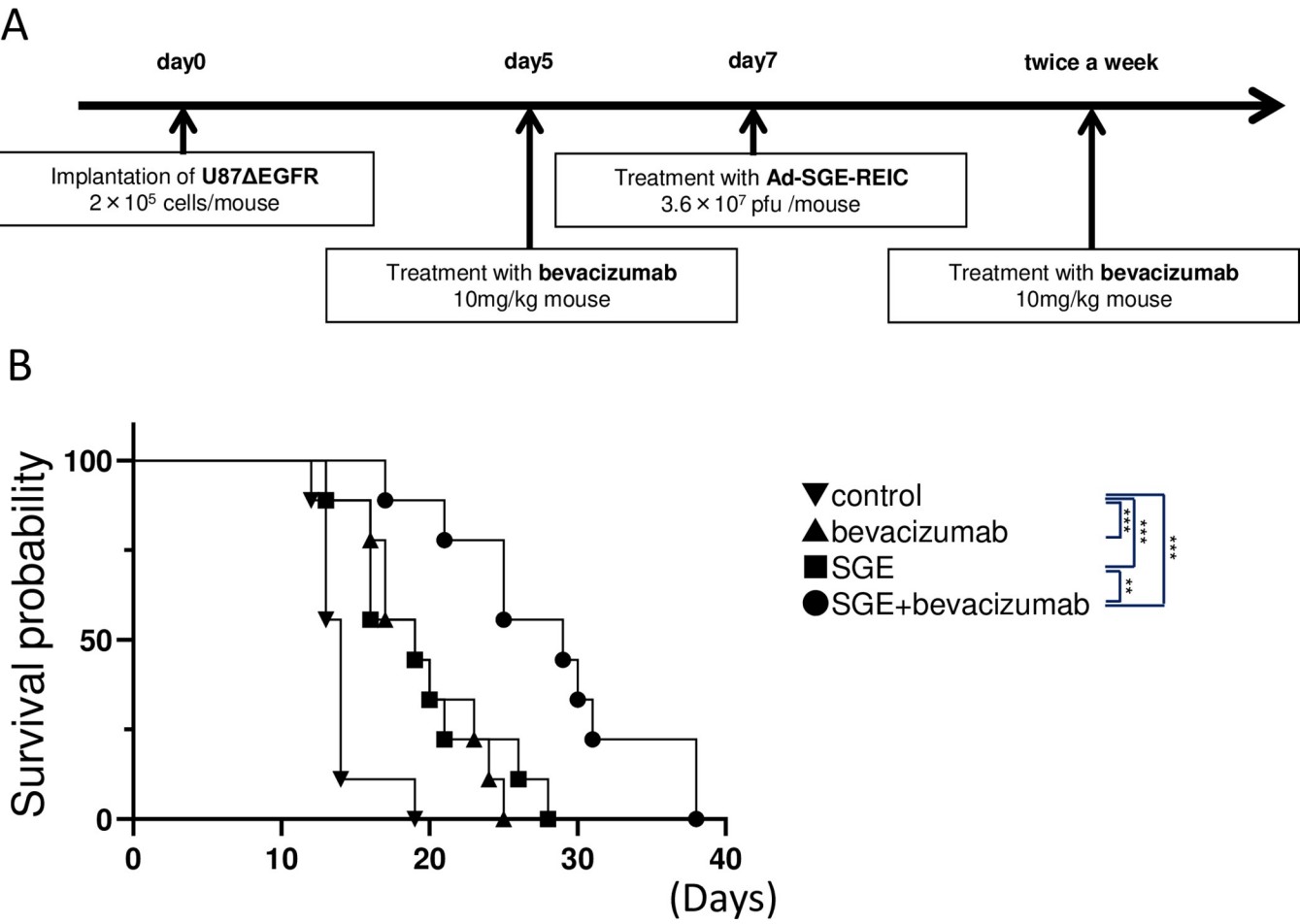

**Fig 4. Kaplan–Meier survival curves of mice implanted with intracranial U87ΔEGFR glioma cells.** (A,B) Glioma cell–bearing animals were administered saline or bevacizumab intraperitonially on the indicated days and intratumoral saline or virus on day 7. Athymic nude mice bearing intracranial U87ΔEGFR gliomas were treated with $3.6 \times 10^7$ pfu Ad-SGE-REIC, and bevacizumab was administered intraperitoneally at 10 μg/g (n = 9 each group). Statistical significance was calculated by the log-rank test. $^{**}p < 0.01$ and $^{***}p < 0.005$ compared with the indicated groups.

## Effect of combining Ad-SGE-REIC with bevacizumab on angiogenesis *in vivo*

To understand the underlying mechanism of the prolongation of survival by the combination treatment *in vivo*, we investigated the histological changes in the tumor-bearing brain. Ad-SGE-REIC is associated with angiogenesis, as shown in Fig 2. Our analysis of The Cancer Genome Atlas (TCGA) glioblastoma dataset revealed a positive correlation of REIC/Dkk3 and PDGFB or PECAM1 (CD31), suggesting the role of REIC/Dkk3 in angiogenesis (Fig 5A and 5B). Athymic mice harboring U87ΔEGFR cell–derived brain tumors were sacrificed 18 days after tumor implantation and immunohistochemical staining using anti-human CD31 was performed (Fig 5C). Treatment with bevacizumab significantly decreased vessel density compared with the control ($p < 0.005$) (Fig 5D). In the group treated with Ad-SGE-REIC, vessel density was significantly increased compared with the control. However, bevacizumab treatment combined with Ad-SGE-REIC significantly decreased the vessel density compared with the levels in Ad-SGE-REIC-treated mice ($p < 0.005$).

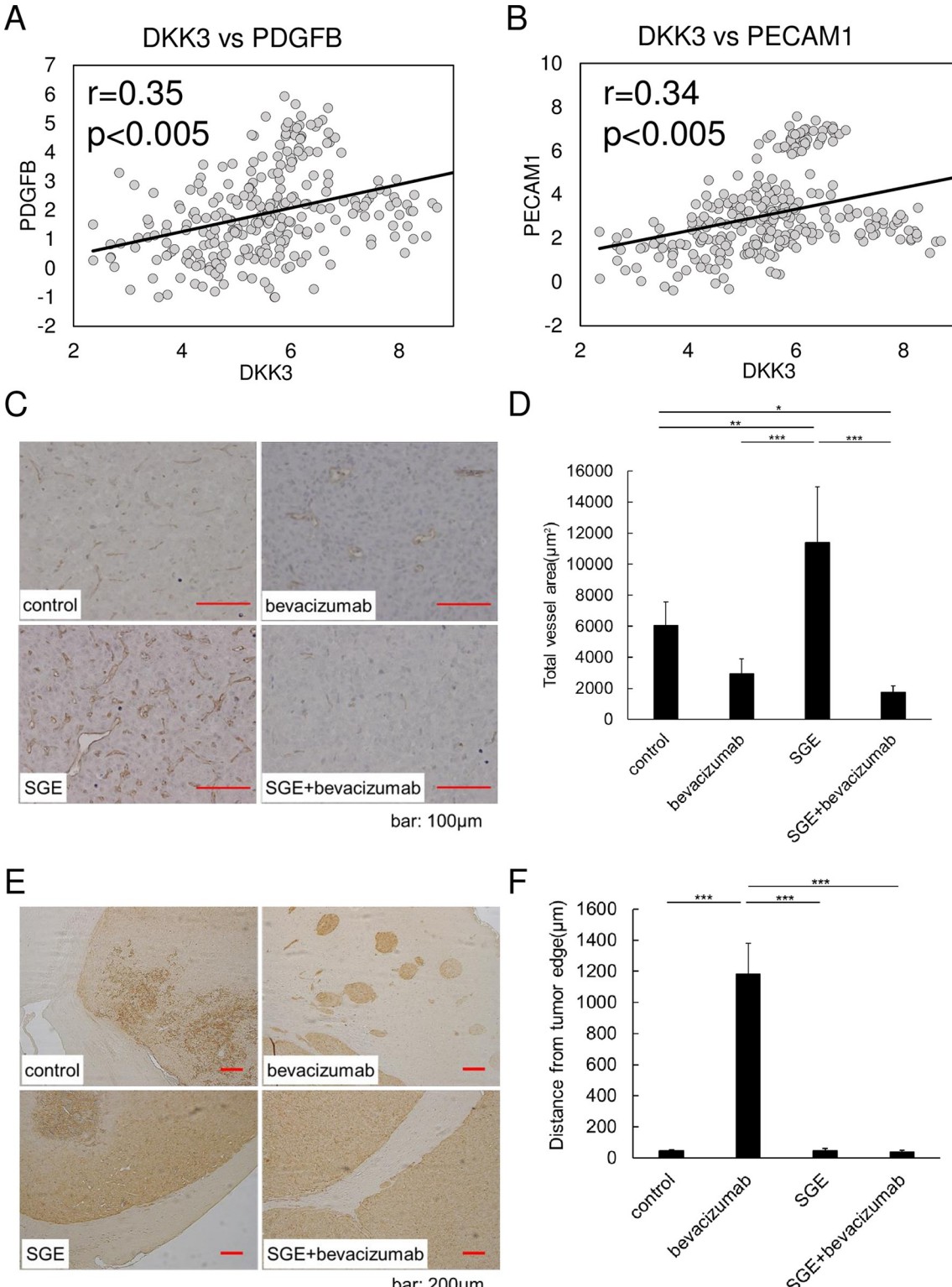

**Fig 5. Anti-angiogenic effect of bevacizumab and anti-invasiveness effect of Ad-SGE-REIC.** (A–B) Analysis of TCGA GBM dataset showed the positive correlation between Dkk3 expression and angiogenesis markers. (C–D) Tumor vessels were identified with CD31. Untreated U87ΔEGFR orthotopic tumor was observed with angiogenic growth. Bevacizumab significantly decreased the vessel density of tumors; the vessel density of the Ad-SGE-REIC-treated group was significantly increased compared with the control. In the combination treatment of Ad-SGE-REIC with bevacizumab, the vessel density was significantly decreased compared with Ad-

SGE-REIC single treatment. (E–F) Immunohistochemical staining of the tumors with anti-human leukocyte antigen monoclonal antibody. The untreated tumor shows the expansion of the tumor with well-defined borders. After treatment with bevacizumab, the tumor border became irregular with tumor invasion. The invasiveness was assessed by the distance between the tumor mass edge and the invasive lesion. Values are the mean ± standard deviation (SD) from five independent experiments. Statistical significance was calculated by one-way of variance (ANOVA) with Tukey's *post hoc* test. $^*p < 0.05$, $^{**}p < 0.01$, and $^{***}p < 0.005$ compared with the indicated groups.

## Effect of Ad-SGE-REIC on bevacizumab-induced invasion *in vivo*

To investigate the effect of Ad-SGE-REIC on bevacizumab-induced glioma cell invasion, immunodeficient mice harboring U87ΔEGFR glioma cells were sacrificed 18 days after tumor implantation, and immunohistochemical staining with anti-human leukocyte antigen was performed. U87ΔEGFR cells treated with bevacizumab alone showed greater invasion to the ipsilateral cerebral cortex adjacent to the injection site compared with the saline controls, Ad-SGE-REIC treated group, or combined bevacizumab and Ad-SGE-REIC treated group (Fig 5E, S1 Fig). Invasion activity was assessed by the distance of invasive tumor lesions from the tumor mass edge. We observed a significant increase in invading glioma cells in the bevacizumab-treated U87ΔEGFR cell group compared with the saline controls ($p < 0.05$). However, combination therapy with bevacizumab and Ad-SGE-REIC significantly decreased the depth of glioma invasion induced by bevacizumab (ipsilateral cortex: $p < 0.05$, Fig 5F). These results demonstrated that Ad-SGE-REIC reduced bevacizumab-induced invasion of glioma cells.

## Microarray analysis of the effect of combination treatment on the U87ΔEGFR orthotopic mouse model

To examine the tumor microenvironment response to the combination therapy, we analyzed the changes in gene expression in tumor tissues from the U87ΔEGFR orthotopic mouse model treated with Ad-SGE-REIC and bevacizumab combination therapy compared with tissues from mice treated with Ad-SGE-REIC monotherapy. We identified 937 upregulated genes and 2565 downregulated genes in Ad-SGE-REIC-treated U87ΔEGFR glioma brain tissue compared with control glioma tissue. We also identified 934 upregulated genes and 397 downregulated genes in the combination treatment group compared with the Ad-SGE-REIC-treated U87ΔEGFR glioma tissue.

**Table 1. Pathway statistics: G1_SGE_vs_control, fold change > 4.**

| Pathway | Criterion for Z score [z] | Permuted P-Value [p] |
|---|---|---|
| Notch Signaling Pathway | 3.23 | 0.004 |
| Dopaminergic Neurogenesis | 2.97 | 0.008 |
| IL-2 Signaling Pathway | 2.8 | 0.009 |
| GPCRs, Class C Metabotropic glutamate, pheromone | 2.7 | 0.013 |
| Adipogenesis genes | 2.61 | 0.01 |
| TYROBP Causal Network | 2.52 | 0.019 |
| SIDS Susceptibility Pathways | 2.37 | 0.017 |
| Retinol metabolism | 2.35 | 0.011 |
| Neural Crest Differentiation | 2.32 | 0.009 |
| Leptin Insulin Overlap | 2.3 | 0.017 |
| TNF-alpha NF-kB Signaling Pathway | 2.11 | 0.035 |
| Delta-Notch Signaling Pathway | 1.99 | 0.034 |
| Heart Development | 1.97 | 0.039 |
| IL-7 Signaling Pathway | 1.97 | 0.032 |

**Table 2. Pathway statistics: G1_SGE_vs_control, fold change < 0.25.**

| Pathway | Criterion for Z score [z] | Permuted P-Value [p] |
|---|---|---|
| Cytoplasmic Ribosomal Proteins | 11.93 | 0 |
| mRNA processing | 9.17 | 0 |
| Electron Transport Chain | 7.43 | 0 |
| Translation Factors | 6.4 | 0 |
| Oxidative phosphorylation | 5.3 | 0 |
| Proteasome Degradation | 5.16 | 0 |
| Exercise-induced Circadian Regulation | 4.46 | 0 |
| Splicing factor NOVA regulated synaptic proteins | 4.43 | 0 |
| Glycolysis and Gluconeogenesis | 3.82 | 0 |
| TNF-alpha NF-kB Signaling Pathway | 3.79 | 0 |
| Mitochondrial Gene Expression | 3.56 | 0 |
| TCA Cycle | 3.51 | 0 |
| Wnt Signaling Pathway NetPath | 3.48 | 0 |
| G Protein Signaling Pathways | 3.45 | 0 |
| Myometrial Relaxation and Contraction Pathways | 3.28 | 0.002 |
| EGFR1 Signaling Pathway | 2.92 | 0.001 |
| G13 Signaling Pathway | 2.48 | 0.012 |
| Calcium Regulation in the Cardiac Cell | 2.46 | 0.012 |
| Hypothetical Network for Drug Addiction | 2.29 | 0.017 |
| IL-3 Signaling Pathway | 2.13 | 0.029 |
| MAPK signaling pathway | 2.03 | 0.041 |
| NLR Proteins | 1.94 | 0.034 |
| Oxidative Stress | 1.9 | 0.046 |
| MAPK Cascade | 1.9 | 0.047 |

Next, we characterized the functional significance of these dysregulated genes using pathway analysis. For the upregulated genes in the Ad-SGE-REIC-treated U87ΔEGFR glioma brain tissue compared with the control, 14 significantly enriched pathways were identified, including the Notch Signaling Pathway, IL-2 Signaling Pathway, and TNF-alpha NF-kB Signaling Pathway (Table 1). For the downregulated genes in Ad-SGE-REIC-treated U87ΔEGFR glioma brain tissue compared with the control, 24 significantly enriched pathways were identified, including the TNF-alpha NF-kB Signaling Pathway, Mitochondrial Gene Expression, and Wnt Signaling Pathway NetPath (Table 2). For the upregulated genes in the combination treated tissue compared with the Ad-SGE-REIC-treated U87ΔEGFR glioma tissue, 12 significantly enriched pathways were identified, including the TNF-alpha NF-kB Signaling Pathway and Delta-Notch Signaling Pathway (Table 3). For the downregulated genes in the combination treated tissue compared with the Ad-SGE-REIC-treated U87ΔEGFR glioma tissue, seven significantly enriched pathways were identified, including the Robo4 and VEGF Signaling Pathways Crosstalk and TNF-alpha NF-kB Signaling Pathway (Table 4).

## Discussion

Our results indicate that the combination therapy of bevacizumab with Ad-SGE-REIC had additional therapeutic effects on glioma cells compared with monotherapy using bevacizumab or Ad-SGE-REIC. Western blot analyses also showed increased expression of several ER stress markers in cells treated with both bevacizumab and Ad-SGE-REIC. Additionally, β-catenin protein levels were potently decreased by combination therapy. In malignant glioma mouse

**Table 3. Pathway statistics: G3_SGE+BEV_vs_SGE, fold change > 4.**

| Pathway | Criterion for Z score [z] | Permuted P-Value [p] |
|---|---|---|
| Cytoplasmic Ribosomal Proteins | 17.89 | 0 |
| mRNA processing | 8.25 | 0 |
| Translation Factors | 7.34 | 0 |
| Exercise-induced Circadian Regulation | 5.17 | 0 |
| Proteasome Degradation | 4.96 | 0 |
| TNF-alpha NF-kB Signaling Pathway | 3.76 | 0 |
| Electron Transport Chain | 3.24 | 0 |
| Oxidative phosphorylation | 2.78 | 0.007 |
| Glycogen Metabolism | 2.48 | 0.013 |
| Hypertrophy Model | 2.38 | 0.018 |
| Delta-Notch Signaling Pathway | 2.23 | 0.02 |
| PluriNetWork | 2.14 | 0.044 |

models, overall survival was extended in the combination therapy group. Our results showed that the invasive activity increased by bevacizumab counteracted the effectiveness of bevacizumab. However, our experiments using the xenograft mouse glioma model indicated that Ad-SGE-REIC inhibited the glioma cell invasion induced by bevacizumab, resulting in a enhanced effect.

## Ad-SGE-REIC

The Ad-SGE-REIC adenovirus vector was developed to increase REIC/Dkk-3 expression and showed enhanced therapeutic effects. We previously demonstrated that Ad-SGE-REIC exhibited time-dependent and significant effects on reducing the number of viable malignant glioma cells in cytotoxicity assays [7]. Xenograft and syngeneic mouse intracranial glioma models treated with Ad-SGE-REIC had significantly longer survival than those treated with the control Ad-LacZ vector or Ad-CAG-REIC [7].

## Combination effect of Ad-SGE-REIC with bevacizumab

In our previous study, we observed a decreased number of vessels in the tumor xenograft model treated with bevacizumab [22], and we demonstrated the prolonged survival of mice treated with bevacizumab. The VEGF autocrine signaling loop is suppressed, the Akt and Erk pathways are activated, and tumor growth and invasion are stimulated by anti-VEGF therapy [26]. Molecules within the extracellular matrix microenvironment, such as proteoglycans and collagens, may influence tumor invasion during anti-VEGF therapy [27]. Our results showed that REIC could inhibit glioma invasion induced by bevacizumab *in vitro* and *in vivo*.

**Table 4. Pathway statistics: G3_SGE+BEV_vs_SGE, fold change <0.25.**

| Pathway | Criterion for Z score [z] | Permuted P-Value [p] |
|---|---|---|
| Glucocorticoid & Mineralcorticoid Metabolism | 3.33 | 0.006 |
| Polyol pathway | 3.18 | 0.023 |
| Robo4 and VEGF Signaling Pathways Crosstalk | 2.48 | 0.036 |
| TNF-alpha NF-kB Signaling Pathway | 2.19 | 0.012 |
| ACE Inhibitor Pathway | 2.04 | 0.022 |
| TGF Beta Signaling Pathway | 1.84 | 0.061 |
| Ptf1a related regulatory pathway | 1.61 | 0.04 |

ER stress markers were upregulated, and, in contrast, β-catenin in the nucleus was downregulated in response to the combination treatment. These results correlated with the anti-invasive effects, which were associated with IRE1α endoribonuclease activity [28, 29]. Microarray data revealed the significantly enriched pathways, including the TNF-alpha NF-kB Signaling Pathway, in the combination treatment sample compared with the Ad-SGE-REIC alone sample. The TNF-alpha NF-kB Signaling Pathway is upstream of ER stress [30]. Future research is needed to investigate this upstream pathway.

Wnt/β-catenin signaling plays an essential role in cellular proliferation, migration, invasion, and angiogenesis, and therefore contributes to glioma progression [31]. Notch1 promotes glioma cell migration and invasion by stimulating β-catenin [32]. Microarray data revealed significantly enriched pathways, including the Notch Signaling Pathway and Delta-Notch Signaling Pathway, in the combination treatment sample compared with the Ad-SGE-REIC alone sample.

## Anti-angiogenic effect of bevacizumab on angiogenesis after REIC treatment

REIC was reported to induce angiogenesis. Untergasser reported that Dkk-3 is expressed in tumor endothelial cells and supports capillary formation [10]. Dkk-3 was also reported to upregulate VEGF in cultured human endothelial cells by activating the activin receptor-like kinase 1 pathway [11]. Our *in vitro* and *in vivo* angiogenesis evaluations revealed significant differences in angiogenesis between the control and Ad-SGE-REIC groups. This might involve the induction of angiogenesis in brain tumors by REIC protein. However, the decrease in angiogenesis with bevacizumab absolutely supported the use of Ad-SGE-REIC treatment (S2 Fig).

## Future directions

Ad-REIC is currently being evaluated in clinical studies. The first in-human, phase I/IIa study of *in situ* Ad-REIC gene therapy for prostate cancer and a phase I/II clinical trial of Ad-SGE-REIC for malignant mesothelioma were performed in Japan [33, 34]. Recently, the safety and efficacy of Ad-SGE-REIC on liver tumors in patients was evaluated in a phase I/Ib study [35]. We started a phase I/IIa clinical trial of Ad-SGE-REIC for the treatment of recurrent malignant glioma, reviewed by the institutional review board (IRB) in March 2019. This is an open-label, single-armed, phase I/IIa study [9].

We previously showed that the integrin antagonist cilengitide improved the effect of Ad-REIC gene therapy for malignant glioma [36]. In addition, Oka et al. reported that immune cells infiltrated tumors treated with Ad-SGE-REIC [7]. We plan to evaluate other combination therapies of Ad-SGE-REIC and molecular targeted therapy, as well as immunotherapy by using syngeneic model. Moreover, U87ΔEGFR cells formed non-invasive tumors into the mouse brain. In the future we need to evaluate patient-derived invasive glioma models.

## Conclusion

We demonstrated the anti-glioma effect of Ad-SGE-REIC. Moreover, our results suggest that the combination therapy of Ad-SGE-REIC and bevacizumab exhibits anti-glioma effects by upregulating the ER stress pathway, downregulating the Wnt signaling pathway, and suppressing the angiogenesis and invasion of tumor cells. Combined Ad-SGE-REIC and bevacizumab may indicate a promising strategy for the treatment of malignant glioma. A phase I/IIa clinical trial of Ad-SGE-REIC for the treatment of recurrent malignant glioma is currently underway.

## Supporting information

**S1 Fig. Low power field of brain tumor immunohistochemically stained with anti-human leukocyte antigen monoclonal antibody.** Marked area with black square is shown in Fig 5E.
(TIF)

**S2 Fig. Tumor invasion and angiogenesis in response to the combination therapy of Ad-SGE-REIC with bevacizumab.** SGE reduces bevacizumab-induced invasion, and bevacizumab suppresses SGE-related angiogenesis.
(TIF)

**S3 Fig. The band densities of protein levels in each group in Fig 3.** We evaluated the band densities of protein levels in each group using Image J (ver. 1.53r). (A) The relative protein expression of Bip and phosphorylated IRE1α were increased in the combination group compared with levels in the individual treatment groups. (B) Downregulation of β-catenin was observed in the combination therapy group compared with other treatment groups in each cell line.
(TIF)

**S1 File. Full scans of the immunoblots shown in the Fig 3.** Boxes indicated parts used in the figure.
(PDF)

## Acknowledgments

We would like to thank M. Arao and Y. Ukai for their technical assistance. N. Akura and N. Yamamoto, medical students at our institution, also contributed to the animal experiments. Bevacizumab was generously provided by Genentech/Roche/Chugai Pharmaceutical Co. We thank Gabrielle White Wolf, PhD, from Edanz Group (www.edanzediting.com/ac) and Chris Rowthorn of Eibunkousei Company (https://www.eibunkousei.net/) for editing a draft of this manuscript.

## Author Contributions

**Conceptualization:** Yasuhiko Hattori, Kazuhiko Kurozumi.

**Data curation:** Yasuhiko Hattori, Kazuhiko Kurozumi, Nobushige Tsuboi.

**Formal analysis:** Yasuhiko Hattori.

**Funding acquisition:** Kazuhiko Kurozumi.

**Investigation:** Yasuhiko Hattori, Keigo Makino, Shuichiro Hirano.

**Methodology:** Yasuhiko Hattori, Kazuhiko Kurozumi, Yoshihiro Otani, Atsuhito Uneda.

**Project administration:** Yasuhiko Hattori, Tetsuo Oka.

**Supervision:** Kazuhiko Kurozumi, Kentaro Fujii, Tetsuo Oka, Hiromi Kumon, Isao Date.

**Validation:** Kazuhiko Kurozumi.

**Writing – original draft:** Yasuhiko Hattori.

**Writing – review & editing:** Yasuhiko Hattori, Kazuhiko Kurozumi, Yoshihiro Otani, Atsuhito Uneda, Yusuke Tomita, Yuji Matsumoto, Yosuke Shimazu, Hiroyuki Michiue, Hiromi Kumon, Isao Date.

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
