## [Decision Letter · Decision Letter 0]

2 Mar 2022

PONE-D-22-02159Combination of Ad-SGE-REIC and bevacizumab modulates glioma progression by suppressing tumor invasion and angiogenesisPLOS ONE

Dear Dr. Kazuhiko Kurozumi,

Thank you for submitting your manuscript to PLOS ONE. After careful consideration, we feel that it has merit but does not fully meet PLOS ONE’s publication criteria as it currently stands. Therefore, we invite you to submit a revised version of the manuscript that addresses the points raised during the review process.

Please submit your revised manuscript within 60 days. If you will need more time than this to complete your revisions, please reply to this message or contact the journal office at plosone@plos.org. Please include the following items when submitting your revised manuscript:

We look forward to receiving your revised manuscript.

Kind regards,

Chunming Liu

Academic Editor

PLOS ONE

Journal Requirements:

- https://www.medrxiv.org/content/10.1101/2020.07.09.20150136v1.full

In your revision ensure you cite all your sources (including your own works), and quote or rephrase any duplicated text outside the methods section. Further consideration is dependent on these concerns being addressed.

4. To comply with PLOS ONE submissions requirements, in your Methods section, please provide additional information on the animal research and ensure you have included details on (1) methods of sacrifice, (2) methods of anesthesia and/or analgesia, (3) efforts to alleviate suffering, (4) basic housing and health monitoring.

5. As part of your revision, please complete and submit a copy of the Full ARRIVE 2.0 Guidelines checklist, a document that aims to improve experimental reporting and reproducibility of animal studies for purposes of post-publication data analysis and reproducibility: https://arriveguidelines.org/sites/arrive/files/Author%20Checklist%20-%20Full.pdf (PDF). Please include your completed checklist as a Supporting Information file. Note that if your paper is accepted for publication, this checklist will be published as part of your article.

6. We suggest you thoroughly copyedit your manuscript for language usage, spelling, and grammar. If you do not know anyone who can help you do this, you may wish to consider employing a professional scientific editing service. 

7. Thank you for stating the following in the Acknowledgments Section of your manuscript: 

"This study was supported by grants-in-aid for Scientific Research from the Japanese Ministry of Education, Culture, Sports, Science and Technology to K. Kurozumi (nos. 26462182 and 17K10865). Our department receives research funding from Momotaro-Gene Co. Ltd., but this fact will not influence the trial results. Okayama University and Momotaro-Gene Inc., a start-up biotech company from Okayama University, hold the patents for the Ad-REIC agent. They are working together on the development of the Ad-REIC agent as a cancer therapeutic medicine."

"This study was supported by grants-in-aid for Scientific Research from the Japanese Ministry of Education, Culture, Sports, Science and Technology to K. Kurozumi (nos. 26462182 and 17K10865)."

8. Thank you for stating the following in your Competing Interests section:  

"H. Kumon is the Chief Scientific Officer of Momotaro-Gene Inc. He demonstrated the utility of the agent and also owns stocks in Momotaro-Gene Inc. The other authors have no other relevant affiliations or financial involvement with any organization or entity with a financial interest in or financial conflict with the subject matter or materials discussed in the manuscript apart from those disclosed."

9. In your Data Availability statement, you have not specified where the minimal data set underlying the results described in your manuscript can be found. PLOS defines a study's minimal data set as the underlying data used to reach the conclusions drawn in the manuscript and any additional data required to replicate the reported study findings in their entirety. All PLOS journals require that the minimal data set be made fully available. For more information about our data policy, please see http://journals.plos.org/plosone/s/data-availability.

10. PLOS ONE now requires that authors provide the original uncropped and unadjusted images underlying all blot or gel results reported in a submission’s figures or Supporting Information files. This policy and the journal’s other requirements for blot/gel reporting and figure preparation are described in detail at https://journals.plos.org/plosone/s/figures#loc-blot-and-gel-reporting-requirements and https://journals.plos.org/plosone/s/figures#loc-preparing-figures-from-image-files. When you submit your revised manuscript, please ensure that your figures adhere fully to these guidelines and provide the original underlying images for all blot or gel data reported in your submission. See the following link for instructions on providing the original image data: https://journals.plos.org/plosone/s/figures#loc-original-images-for-blots-and-gels. 

Reviewers' comments:

Reviewer's Responses to Questions

**Comments to the Author**

1. Is the manuscript technically sound, and do the data support the conclusions?

Reviewer #1: Yes

Reviewer #2: Yes

2. Has the statistical analysis been performed appropriately and rigorously? 

Reviewer #1: Yes

Reviewer #2: Yes

3. Have the authors made all data underlying the findings in their manuscript fully available?

Reviewer #1: Yes

Reviewer #2: Yes

4. Is the manuscript presented in an intelligible fashion and written in standard English?

Reviewer #1: Yes

Reviewer #2: Yes

5. Review Comments to the Author

Reviewer #1: In this experimental study the authors evaluate the combination treatment of Ad-SGE-REIC and bevacizumab on glioma. The authors conclude that their results demonstrate that the combination therapy of Ad-SGE-REIC and bevacizumab exhibits anti-glioma effects by upregulating the ER stress pathway, downregulating the Wnt signaling pathway, and suppressing the angiogenesis and invasion of tumor cells.

The therapeutic hypothesis proposed by the authors is very interesting.

The authors should better specify the mechanism of β-catenin reduction induced by the drug compound.

However, further studies should be carried out to confirm these results.

Reviewer #2: In this manuscript the authors evaluated the potential anti-glioma effect of a combination therapy using DKK-3 overexpressing Ad-SGE-REIC and VEGF antibody bevacizumab. After reading the manuscript, I would recommend accepting it after minor revisions.

1), In figure 1, the legends of panels C, D and E, F were inverted.

2), In figure 3A, the authors claimed significance between the combination group and individual treatments while those bands were not visually different. In figure 3B, the loading control TBP for U87�EGFR for the combination group was about half of the other groups, and thus it is unsupported to conclude that beta-catenin was down regulated.

3) Why was the advantage of using EGFR deleted U87 cells instead of regular U87?

6. PLOS authors have the option to publish the peer review history of their article (what does this mean?). If published, this will include your full peer review and any attached files.

Reviewer #1: **Yes: **Gerardo Caruso

Reviewer #2: No

---

## [Author Response · Author response to Decision Letter 0]

30 Jul 2022

Response to Reviewers

We would like to express our appreciation to the reviewers for their insightful comments, which helped us significantly improve our paper.

#1 The authors should better specify the mechanism of β-catenin reduction induced by the drug compound.

Response:

Regarding the mechanism of β-catenin reduction, Behzadian reported that anti-VEGF therapy causes reduction of uPAR gene expression by transcriptional activation of β-catenin [1]. In addition, Moreau reported that β-catenin and NF-κB cooperate to regulate the uPA/uPAR system in cancer cells [2]. We performed preliminary western blot analysis and observed decreased uPAR protein expression in cells treated by the combination treatment (as shown in Response to Reviewers). Future studies are required to determine the precise mechanism.

References

1. M. Ali Behzadian, L. Jack Windsor, Nagla Ghaly, Ruth B. Caldwell_et_al. VEGF-induced paracellular permeability in cultured endothelial cells involves urokinase and its receptor. FASEB J. 2003 Apr;17(6):752-4. doi: 10.1096/fj.02-0484fje. Epub 2003 Feb 19. PMID: 12594181

2. Moreau M, Mourah S, Dosquet C. β-Catenin and NF-κB cooperate to regulate the

uPA/uPAR system in cancer cells. Int J Cancer. 2011 Mar 15;128(6):1280-92. doi:

10.1002/ijc.25455. PMID: 20473943.

#2 In figure 1, the legends of panels C, D and E, F were inverted.

Response:

We have corrected this information in the revised manuscript.

#3 In figure 3A, the authors claimed significance between the combination group and individual treatments while those bands were not visually different. In figure 3B, the loading control TBP for U87DEGFR for the combination group was about half of the other groups, and thus it is unsupported to conclude that beta-catenin was down regulated.

Response: 

We appreciate the insightful observation. We quantified the band densities of protein expression of Bip and phosphorylated IRE1α between the combination group and individual treatments using Image J (ver. 1.53r). We also showed the downregulation of β-catenin in response to the combination therapy compared with other treatment groups in each cell line. We added these data in supporting figure 3.

#4 Why was the advantage of using EGFR deleted U87 cells instead of regular U87?

Response:

We would like to show the anti-glioma effects using glioma cell lines with high malignancy and U87ΔEGFR is one of the aggressive glioma cell lines. Moreover, TCGA database shows that glioblastoma patients have EGFR mutation or amplification, and therefore we considered that data in U87ΔEGFR cells may be more applicable to the clinic than data in U87 cells. We also reported the anti-glioma effects of Ad-SGE-REIC in U87ΔEGFR cells, and we would compare the higher effectiveness of combination therapy to Ad-SGE-REIC alone with the same cell line.

TCGA data is attached in "Response to Reviewers".

In addition

We added 2 authors to this study, because they helped our revise research.

---

## [Decision Letter · Decision Letter 1]

5 Aug 2022

Combination of Ad-SGE-REIC and bevacizumab modulates glioma progression by suppressing tumor invasion and angiogenesis

PONE-D-22-02159R1

Dear Dr. Kazuhiko Kurozumi,

We’re pleased to inform you that your manuscript has been judged scientifically suitable for publication and will be formally accepted for publication once it meets all outstanding technical requirements.

Kind regards,

Chunming Liu

Academic Editor

PLOS ONE

Additional Editor Comments (optional):

Reviewers' comments:

Reviewer's Responses to Questions

**Comments to the Author**

1. If the authors have adequately addressed your comments raised in a previous round of review and you feel that this manuscript is now acceptable for publication, you may indicate that here to bypass the “Comments to the Author” section, enter your conflict of interest statement in the “Confidential to Editor” section, and submit your "Accept" recommendation.

Reviewer #1: All comments have been addressed

Reviewer #2: All comments have been addressed

2. Is the manuscript technically sound, and do the data support the conclusions?

Reviewer #1: (No Response)

Reviewer #2: Yes

3. Has the statistical analysis been performed appropriately and rigorously? 

Reviewer #1: (No Response)

Reviewer #2: Yes

4. Have the authors made all data underlying the findings in their manuscript fully available?

Reviewer #1: (No Response)

Reviewer #2: Yes

5. Is the manuscript presented in an intelligible fashion and written in standard English?

Reviewer #1: (No Response)

Reviewer #2: Yes

6. Review Comments to the Author

Reviewer #1: (No Response)

Reviewer #2: All my questions have been addressed by the authors. I have no other questions. It is ready to accept.

7. PLOS authors have the option to publish the peer review history of their article (what does this mean?). If published, this will include your full peer review and any attached files.

Reviewer #1: **Yes: **Gerardo Caruso

Reviewer #2: No

---

## [Editor Report · Acceptance letter]

16 Aug 2022

PONE-D-22-02159R1 

Combination of Ad-SGE-REIC and bevacizumab modulates glioma progression by suppressing tumor invasion and angiogenesis 

Dear Dr. Kurozumi:

I'm pleased to inform you that your manuscript has been deemed suitable for publication in PLOS ONE. Congratulations! Your manuscript is now with our production department. 

Kind regards, 

on behalf of

Dr. Chunming Liu 

Academic Editor

PLOS ONE